# Removal of Typical PPCPs by Reverse Osmosis Membranes: Optimization of Treatment Process by Factorial Design

**DOI:** 10.3390/membranes13030355

**Published:** 2023-03-20

**Authors:** Jianing Liu, Liang Duan, Qiusheng Gao, Yang Zhao, Fu Gao

**Affiliations:** 1State Key Laboratory of Environmental Criteria and Risk Assessment, Chinese Research Academy of Environmental Sciences, Beijing 100012, China; 2Technical Centre for Soil, Agriculture and Rural Ecology and Environment, Ministry of Ecology and Environment, Beijing 100012, China

**Keywords:** reverse osmosis membrane process, PPCPs, removal efficiency, adsorption model, factorial design

## Abstract

In this paper, the removal effect of reverse osmosis (RO) on three common pharmaceuticals and personal care products (PPCPs), including ibuprofen (IBU), carbamazepine (CBZ), and triclosan (TCS), were compared under different process conditions, and the removal rate of PPCPs, membrane flux, and PPCPs membrane adsorption capacity were analyzed. The removal rate increased with the increase of the influent concentration and pre-membrane pressure, while pH influenced the removal effect of different PPCPs by affecting the electrostatic interaction between pollutants and membranes. It was also found that the dynamic adsorption of PPCPs on RO membranes under different conditions complied with the pseudo-first-order reaction kinetic adsorption model. The maximum stable adsorption capacity and the adsorption rate of PPCPs on membranes under various conditions were simulated based on the model. Moreover, through factorial design, the removal rates of RO on IBU, CBZ, and TCS could reach 98.93%, 97.47%, and 99.01%, respectively, under the optimal conditions (with an influent concentration of 500 μg/L, pre-membrane pressure of 16 bar and pH = 10). By optimizing the process of removing PPCPs with the RO membrane method, the optimal process conditions of removing IBU, CBZ, and TCS with the RO membrane method were obtained, which provided reference conditions and data support for the practical application of removing PPCPs with the RO membrane method.

## 1. Introduction

PPCPs are organic pollutants that are more commonly found in water bodies, and related research shows that traditional water treatment methods have a limited capacity to remove PPCPs [1,2,3]. Reverse osmosis has been widely employed in treating many water bodies due to its efficiency and high selectivity advantages, particularly the effective removal of small molecule micropollutants [4,5,6,7,8]. However, when using RO membranes to remove organic trace pollutants such as low-concentration PPCPs in water bodies, the removal effect usually fails to satisfy the expectations [9,10]. RO membrane rejection mechanisms on PPCPs are relatively complex and closely related to the physical and chemical properties of membranes and pollutants and membrane system operation. In addition, existing research mainly focuses on the influence factors and removal efficiency of the removal of PPCPs by RO membranes [5,11], while there is only a little research regarding process optimization. Therefore, in order to improve the water treatment effect of RO membranes, it is necessary to clarify the rejection effect of RO membranes on PPCPs and optimize the treatment process conditions.

At present, size repulsion, electrostatic repulsion, and adsorption are the primary mechanisms by which RO membranes reject organic micropollutants. It was found that the molecular weight index could not accurately measure the removal efficiency of substances during size repulsion [12,13] due to the differences in molecular length and shape [14,15]; using Stokes molecular radius as the size parameter can only slightly improve the correction of removal rate [16]. The rejection effect of size repulsion on pollutants can be explained more accurately only by understanding pollutant molecules’ spatial shape and membrane pore structure. In addition, the charge of the membrane surface and the substance can be changed by pH, and the rejection performance of pollutants can be affected through the electrostatic interaction of different intensities [17,18]. Additionally, pH can affect the filtration performance of membranes by changing the extended chain structure formed by charged groups in membrane pores [19] or result in hydrolysis of chemical bonds and reduce the degree of crosslinking [20]. Moreover, membrane material has an adsorption effect on some organic micropollutants [21,22], with hydrogen bonding, π-π bonding, hydrophobic interaction, and electrostatic interaction being the main adsorption mechanism of pollutants and membranes [23,24,25]. Membrane adsorption happens on the membrane surface and throughout the entire structural layer. It is influenced by the membrane filtration pressure, membrane material, and pollutant’s physical and chemical properties [26].

With respect to the study, both the removal process of RO membranes on three common PPCPs and the influence factors of RO membranes on PPCPs removal during the rejection were studied. The impacts of influent PPCP concentrations, pre-membrane pressure, and influent pH on the removal of PPCPs were investigated through dynamic experiments. The law of adsorption of PPCPs on RO membranes under different control conditions was analyzed through adsorption experiments. Finally, the factorial design was used to optimize the process and find the optimum operating conditions for PPCP removals by reverse osmosis, to improve the removal effect of common PPCPs.

## 2. Materials and Methods

### 2.1. Experimental Device

#### 2.1.1. Target Pollutants

In the study, three common PPCPs, including ibuprofen (IBU), carbamazepine (CBZ), and triclosan (TCS), were selected as target pollutants. The stock solutions for these three drugs were prepared using chromatographic grade methanol and stored at 4 °C in brown reagent bottles. Three standard substances of target pollutants were purchased from Macklin (Shanghai, China) and Alfa Aesar (Shanghai, China), and the purity of each standard substance was over 98%. The methanol and acetonitrile used for experiments were purchased from Thermo Fisher (Shanghai, China), with HPLC grade or above.

#### 2.1.2. RO Device

The RO membrane used in the study was Dow BW30 reverse osmosis membrane from Dow Chemical Company (Midland, MI, USA). The high-pressure flat membrane equipment was used for the membrane filtration experiment (Figure 1), and the area of the membrane cell of the device was 24 cm^2^. Besides, multiple types of diaphragms could be configured, and the range of system filtration pressure was 1–60 bar, with a temperature range of 0–80 °C and a filtration capacity of 0–1 L/Hr. Moreover, the water inlet temperature was controlled through the thermostatic water tank, and the penetrating fluid was collected. The effluent characteristics were continuously collected by the electronic scale to calculate the permeation flux. The concentrated water flowed back to the feed water reservoir.

Before starting the experiment, the RO membranes used were soaked in ultrapure water for at least 24 h, cut into the appropriate size, and put into the filter membrane cell. Under a certain pressure and temperature of 25 °C, the ultrapure water was run in the system for 6 h until the water flux reached a stable level to eliminate the impact of membranes in the process of compaction.

### 2.2. Experimental Process Design

#### 2.2.1. PPCP Removal Experiment

A certain quantity of 100 mg/L PPCPs mixed standard solution was added to bring the experimental solution to the set concentration, and the background ion concentration was controlled by adding NaCl solution. The pH was adjusted by 1 M HCl and NaOH. In addition, the computer was connected to the electronic scale under the permeation collection bottle, and the effluent change was recorded at a time interval of 1 min in real time to obtain the water flux data. The recovery rate of penetrating fluid in the reverse osmosis experiment was 70%. The raw water in the feed water reservoir was sampled before the experiment. The penetrating fluid and residual water in the feed water reservoir were collected after the experiment, and the concentration of PPCPs in three water bodies was measured to analyze the change in PPCP concentrations.

The impact of PPCP concentrations of inlet water (50 μg/L, 100 μg/L, 500 μg/L, and 1000 μg/L), pre-membrane pressure (8 bar, 12 bar, 16 bar, and 20 bar), and pH of inlet water (3, 5, 7, 9, and 11) on the removal rate of PPCPs in reverse osmosis experiment were studied respectively.

#### 2.2.2. PPCPs Dynamic Adsorption Experiment

The same experimental devices were used in the dynamic adsorption experiment, with the exception that effluent penetrating fluid recirculated back the feed water reservoir for the continuous experiment in the reverse osmosis experiment. In addition, the set runtime was 6 h, and the feed water reservoir was sampled every 30 min in the first 3 h and 1 h in the last 3 h. The samples were collected in brown liquid sample bottles for measurement after being filtered by a 0.22 μm filter membrane. Moreover, the change in PPCP concentrations of inlet water was measured during the experiment to analyze the law of adsorption of RO membranes on PPCPs.

The impact of PPCP concentrations of inlet water (50 μg/L, 100 μg/L, 500 μg/L, and 1000 μg/L), pre-membrane pressure (8 bar, 12 bar, 16 bar, and 20 bar), and pH of inlet water (3, 5, 7, 9, and 11) on the adsorption of PPCPs on membranes in reverse osmosis experiment were studied respectively.

### 2.3. Analysis and Determination Method

#### 2.3.1. PPCPs Concentration Determination

Raw water, osmotic solution, and residual water were collected and filtered by a 0.45 μm filter membrane to remove suspended particles. An HLB column was used for SPE water sample pretreatment. The samples were determined by Agilent 1260 high-performance liquid chromatography (Santa Clara, CA, USA). The chromatographic column was ZORBAX Eclipse XDB-C18. The mobile phase consisted of acetonitrile and ultra-pure water with a pH of 3 regulated by phosphoric acid.

#### 2.3.2. Calculation of Permeation Flux and PPCPs Dynamic Adsorption Amount

In this experiment, an electronic scale was used to record the effluent characteristics, and the flux data of the RO membrane was represented by the mass change. In order to eliminate the influence of different diaphragms and other factors in each experiment, the ratio (J/J_0_) of experimental solution flux data (J) and real-time stable flux data of ultra-pure water pressure (J_0_) were selected as analysis objects.

The adsorption amount Q (μg/cm^2^) of PPCPs on the membrane is calculated based on the following formula:Q = ((C_0_ − C_e_)·V)/A,(1)C_0_ and C_e_ are the concentration of PPCPs (μg/L) in the feed water reservoir at initial and stable adsorption equilibrium, respectively. V indicates the volume of the solution (L), and A represents the effective membrane area (cm^2^).

Assuming that the active centers on the membrane surface are limited and the adsorption can reach an equilibrium state over time, the first-order kinetic model shown below can be used to indicate the adsorption [27]:Q = Q_e_(1 − e^−kt^),(2)Q_e_ is the maximum adsorption mass (μg/cm^2^) at adsorption equilibrium, and k is the rate constant.

### 2.4. Experiment Design and Optimization

To analyze the removal effect of reverse osmosis on common PPCPs, factorial analysis was used to reduce the number of experiments and obtain parameter optimization [28] of influent PPCP concentrations, pre-membrane pressure, and pH value, and analyze the influence of each factor on the reaction as well as interactions between all factors [28,29]. The secondary factor design of Design Expert 8 was used to predict the results. Table 1 shows the factor encoding and the scope of variables.

## 3. Results and Discussion

### 3.1. Influence of Different Conditions on the Removal Rate and Permeation Flux of PPCPs

#### 3.1.1. PPCP Removal Rates

As seen in Figure 2, the removal rates of the three PPCPs were all greater than 94% under different concentrations. The highest removal rate was 98.90% when the initial IBU concentration was 1000 g/L; the lowest rate was 96.93% when the initial IBU concentration was 100 μg/L. The removal rate of CBZ increased and varied between 94.42% and 98.13% when the initial concentration increased. The removal rate of TCS exceeded 98%. In different concentration ranges, the removal rate of IBU and TCS varied within 2%, and the removal rate of CBZ further varied within 4%. It was preliminarily suggested that the removal rate of IBU and TCS was less affected by the influent concentration at μg-level concentration, and the removal rate of CBZ varied more obviously with the initial concentration. Under different pre-membrane pressure conditions, the removal rate of the three PPCPs was the lowest at 8 bar. The removal rate of IBU and CBZ gradually increased as the pressure rose from 8 to 16 bar and slightly decreased when the pressure was 20 bar. TCS removal rate was higher than 99% under different pressure conditions. The lowest removal rate was 96.41% when IBU was at 8 bar, and the removal rate was greater than 98% when the pressure was higher than 12 bar, with less than a 1% difference. The removal rate of CBZ was 93.86%−98.07%. When the pH = 3–11, the removal rate of PPCPs with RO membrane varied from 2% to 4%. The removal rate of TCS at different pH values was greater than 99%, and that of IBU and CBZ decreased first and then increased when the pH = 3–11.

Among the three PPCPs, the molecular weight (289.54) and Stokes molecular radius (0.415) of TCS were the largest, which maintained a high removal rate in reverse osmosis interception that was dominated by size repulsion and was unaffected by pH variation. The removal variation of IBU and CBZ indicated that, in addition to the size repulsion, certain electrostatic repulsions existed during reverse osmosis interception of PPCPs. The pKa value of IBU was 4.5. When pH > pKa, IBU was ionized into anions, which enhanced the repulsion with a negatively charged membrane and increased the removal rate. The pH was the main factor affecting the zeta potential of the membrane [30]. When the pH is above the isoelectric point, the membrane is negatively charged. Otherwise, it is positively charged. Studies have shown that negatively charged organic matter has a significantly higher removal rate than uncharged organic matter [12], regardless of differences in the physicochemical properties of the membrane or physicochemical properties of organic matter. It is mainly because negatively charged organic matter can be subject to electrostatic repulsion with the membrane. After changing the pH of water, the charge on the membrane surface can be effectively changed, affecting the existence form of the organic matter. The removal rate of organic matter can then be affected by the interaction between the membrane and the organic matter [31]. For dense RO membranes, size repulsion is the main mechanism of PPCP removals. Electrostatic repulsion can affect the removal rate by influencing the interaction between PPCPs and the membrane, and the final retention effect is the result of size repulsion and electrostatic repulsion.

#### 3.1.2. Permeation Flux

As seen in Figure 3, membrane flux basically remained stable under different PPCP concentrations, but the stable permeation flux was different. The permeation flux was relatively close when the initial concentrations were 50 μg/L and 100 μg/L, respectively. When the initial concentration was increased to 500 μg/L, the permeation flux decreased by 23.6%; when the initial concentration was 1000 μg/L, the permeation flux decreased by 37.2%. The decrease in membrane flux enhanced the interception effect of PPCPs, and the removal rate of PPCPs increased along with the initial concentration. The pre-membrane pressure and permeation flux remained stable, but the effluent operation time with a fixed recovery rate (70%) differed (8 h−24 h). The stable permeation flux increased as pressure was added. Compared with *p* = 8 bar, the stable permeation flux when *p* = 20 bar increased by 22.2%. Under different pH conditions, the permeation flux gradually declined over time during the experiment. The continuous increase in the concentration of PPCPs in raw water led to a decrease in permeation flux. The stable permeation flux decreased as the pH increased, possibly because the BW30 membrane is made of polyamide. When the pH increased, electrostatic repulsion between carboxyl groups prolonged the chain, thus reducing the effective pore size of the membrane and the stable permeation flux of the RO membrane [32]. Electrostatic interaction caused by pH variation affected the stable flux.

### 3.2. Influence of Different Conditions on the Adsorption of PPCPs on the Membrane

In the dynamic adsorption experiment (as shown in Figure 4), the concentration of the three PPCPs decreased exponentially based on preliminary analysis. As the decrease tended to be stable over time, adsorption equilibrium between membrane and PPCPs was reached. The three PPCPs had different amounts of adsorption on the membrane. The kinetic parameters of the pseudo-first-order reaction model of dynamic adsorption were calculated. The correlation coefficients were relevantly high (Table A1, Table A2 and Table A3 in the Appendix A), indicating that the first-order reaction model could well explain the adsorption of PPCPs on the membrane.

The adsorption of the three PPCPs on the membrane increased as the initial concentration rose. The adsorption rates of the three PPCPs on the membrane were IBU > TCS > CBZ, with the difference in adsorption rate unaffected by the concentration. Under different pressure conditions, the stable adsorption amount of the three PPCPs follows the rules of TCS ≈ IBU > CBZ, and the stable adsorption amount of the three PPCPs decreased as pre-membrane pressure increased. The adsorption rate of CBZ and TCS increased as the pressure went up. The adsorption rate of IBU at low pressure (8 and 12 bar) is higher than that at high pressure (16 and 20 bar). As the pH value increased, the adsorption amount of IBU and TCS decreased significantly, while the adsorption amount of CBZ showed an overall increasing trend, while the variation was not significant. The adsorption amounts follow the rules of TCS > IBU > CBZ.

The hydrophobic adsorption between hydrophobic PPCPs and the membrane can facilitate the retention of PPCPs to a certain extent before the adsorption equilibrium is reached. The adsorption is not only related to the hydrophobicity of PPCPs but also in connection with the physicochemical properties of PPCPs (such as molecular size and selectivity) and membrane characteristics (pore size, charge, and roughness). CBZ (logKow = 2.45) performs worst in hydrophobicity, while BW30 reverse osmosis membrane tends to be hydrophobic. As the adsorption process of PPCPs is lower with relatively weak hydrophobicity, it takes longer for CBZ to reach adsorption equilibrium on the membrane. The logKow values of the three PPCPs followed the rules of TCS (4.8) > IBU (3.97) > CBZ (2.45). The results showed that PPCP adsorptions were driven by the hydrophobic affinity between PPCPs and the membrane surface. However, the adsorption amount of IBU and TCS was lower than that at a low pH value when pH > pKa. Because the pKa of CBZ was larger (13.9), and the adsorption amount was less affected by pH variation, this indicated that the electrostatic repulsion between the membrane and PPCPs also influences the adsorption effect of PPCPs on the membrane surface [33]. Studies suggested that although estradiol has a high hydrophobicity, its adsorption effect with the membrane is lower than expected due to electrostatic repulsion [34]. The adsorption amount is also related to the surface roughness of the membrane. For example, the adsorption amount of a relatively smooth membrane for salicylic acid is 81%, compared to 94% of a rough membrane [34]. When analyzing the adsorption of PPCPs and membranes, multiple factors should be considered comprehensively.

### 3.3. Reverse Osmosis Process Optimization

#### 3.3.1. Factorial Design Experiment

Based on a 2^k^ factorial design, sixteen experimental operations (including two replicates per experiment) were optimized. Table 2 shows the actual percentage of three types of PPCP removals. Randomized experiments were conducted to determine the effects of each factor on the response.

#### 3.3.2. Analysis of Variance (ANOVA)

Analysis of variance was adopted to observe the main and interaction factors affecting PPCP removals. Table 3 shows the variance analysis results for the three responses. The importance of each factor is quantified by the sum of squares (SS), and its importance in the influence also increases as the SS value increases. A *p*-value less than 0.05 was adopted to determine the potential meaning of each main and interaction effect [35].

#### 3.3.3. Main and Interaction Effect

When using the coefficient of determination (R^2^) to measure the proportion of the total variability explained by the model, the *f*- and *p*-values were adopted to calculate the significance of each coefficient, and the coefficient of determination should be at least 0.8 [36]. Table 3 shows that the R^2^ value in the IBU removal experiment was close to 1(0.9294). The model explains the variability of 92.94% in the data, and the *f-* and *p*-values of the model are 15.04 and 0.0005, respectively. Besides, it shows that the model fully describes the experimental data. A (influent IBU concentration) and B (pre-membrane pressure) were the biggest factors affecting the removal of IBU. A (influent CBZ concentration) and B (pre-membrane pressure) were the main factors affecting the removal of CBZ. The secondary factor affecting the removal of CBZ was the interaction effect of A and B and C (pH). The *f*- and *p*-values of the model were 5.20 and 0.0168, respectively, and the R^2^ of the model was 0.8199, which meets the requirement of data description. R^2^ of the TCS removal experiment was 0.9984, which explains 99.84% of the variability in the data, with *f-* and *p*-values of 726.45 and below 0.0001, respectively. Thereof, the A (influent TCS concentration), B (pre-membrane pressure), and interaction of A and B were the main factors affecting TCS removal, followed by C (pH). Figure 5 shows the effects of A, B, and C on the three PPCPs, and after discarding unimportant factors, the coded equation can be obtained as follows:IBU removal (%) = 98.65 + 0.39 × A + 0.94 × B − 0.019 × C − 0.10 × A × B,(3)
CBZ removal (%) = 96.65 + 1.10 × A + 1.06 × B + 0.69 × C − 0.75 × A × B,(4)
TCS removal (%) = 98.96 + 0.44 × A + 0.29 × B − 0.037 × C − 0.13 × A × B,(5)

Figure 5 shows the response surface diagram of the removal rate of three PPCPs: IBU, CBZ, and TCS, and the three variables of concentration, pressure, and pH. It can be seen from the figure that concentration and pre-membrane pressure are the most significant factors affecting the removal rate of PPCPs. The effect of pH on the removal rate is relatively minor. This result is consistent with the conclusion of the variance analysis.

#### 3.3.4. Operation Condition Optimization

The removal effect of the three PPCPs is optimized by multiple response methods of expected (D) functions, including the optimization of process parameters for influent PPCP concentrations, pre-membrane pressure, and pH. The optimal level of the three PPCP removals was found. To achieve the maximum results, considering the actual operating conditions of reverse osmosis, the influent PPCP concentrations range was set to 50–500 μg/L, and the pre-membrane pressure range was set to 8–16 bar, with the pH range set to 4–10. The removal rates of the three PPCPs were set to the maximum (with the removal importance level 5/5 for IBU, 5/5 for CBZ, and 3/5 for TCS). Figure 6 shows the generated numerical optimization diagram: When the initial concentration of PPCPs was 500 μg/L, pre-membrane pressure was 16 bar, with the pH of 10, the three PPCPs, including IBU, CBZ, and TCS had the best removal rates, which were 98.93%, 97.47%, and 99.01%, respectively, and the expected value was 0.667.

## 4. Conclusions

The paper studied the removal effect of reverse osmosis membranes on three common PPCPs under different process conditions. The factorial design (FD) method was adopted to determine the main parameter conditions to study the interaction effect of each condition, and the reverse osmosis process conditions relating to the removal of IBU, CBZ, and TCS were optimized. Analysis of Variance (ANOVA) and F-test were adopted to determine the most important variable conditions.

As shown in the preliminary experiment, the removal rate of PPCPs increased with the increase of the initial influent concentration and the pre-membrane pressure. The pH affected the removal effect of different PPCPs by influencing the electrostatic effect between the pollutants and the membrane, and the final removal effect of PPCPs was the result of the combined action of size repulsion and electrostatic repulsion. Under different conditions, the dynamic adsorption of PPCPs by the RO membrane all met the pseudo-first-order reaction kinetic adsorption model. Through model fitting, it was found that the adsorption capacity of PPCPs on the membrane increased with the increase of initial concentration and decreased with the increase of pressure before the membrane. The hydrophobic affinity between PPCPs and the membrane surface drove the adsorption. However, electrostatic repulsion also affects the adsorption effect of PPCPs on the membrane surface.

When the process was optimized by the factorial design method, it was found that the removal rates of IBU, CBZ, and TCS under the best conditions (with an influent concentration of 500 μg/L, pre-membrane pressure of 16 bar, and pH of 10) were 98.93%, 97.47%, and 99.01%, respectively.

## Figures and Tables

**Figure 1 membranes-13-00355-f001:**
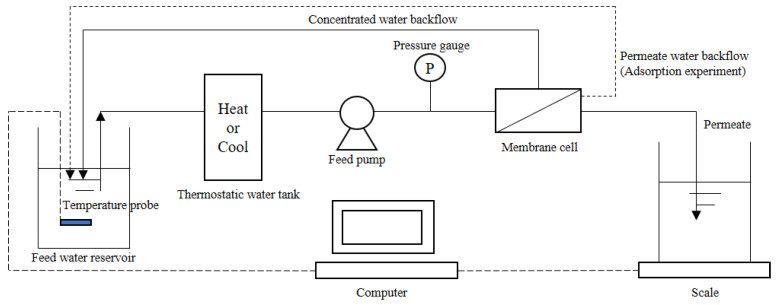
Flow diagram of the reverse osmosis device system.

**Figure 2 membranes-13-00355-f002:**
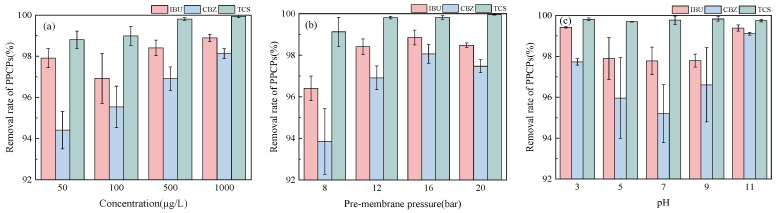
Removal rate variation of various PPCPs under different conditions regarding (**a**) Concentration; (**b**) Pre-membrane pressure, and (**c**) pH.

**Figure 3 membranes-13-00355-f003:**
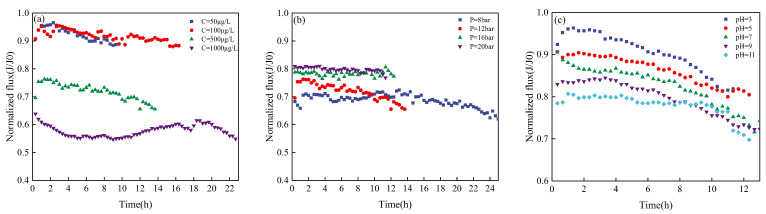
Membrane flux variation of various PPCPs under different conditions regarding (**a**) Concentration; (**b**) Pre-membrane pressure, and (**c**) pH.

**Figure 4 membranes-13-00355-f004:**
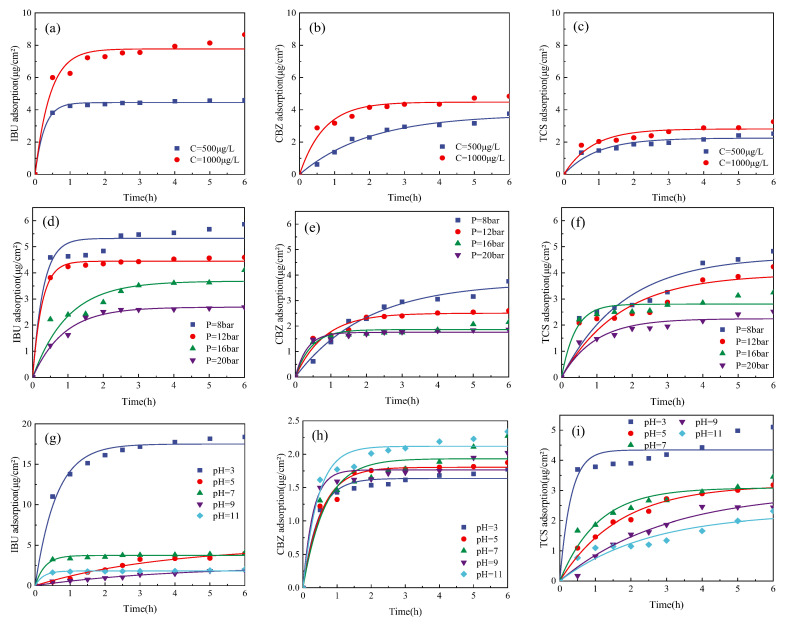
Adsorption kinetics regarding (**a**–**c**) Concentration, (**d**–**f**) Pre-membrane pressure, and (**g**–**i**) pH of various PPCPs under different conditions.

**Figure 5 membranes-13-00355-f005:**
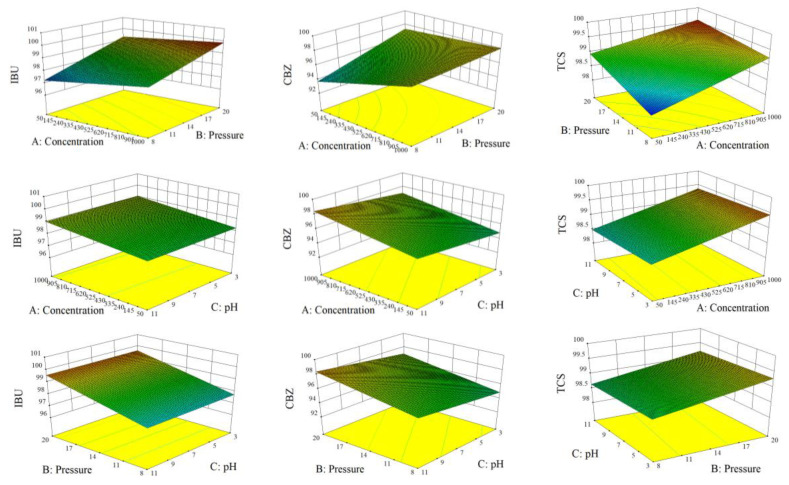
Response surface diagram of three PPCP removal rates. The concentration range is 50–1000 μg/L, the pressure range is 8–20 bar, and the pH range is 3–11.

**Figure 6 membranes-13-00355-f006:**
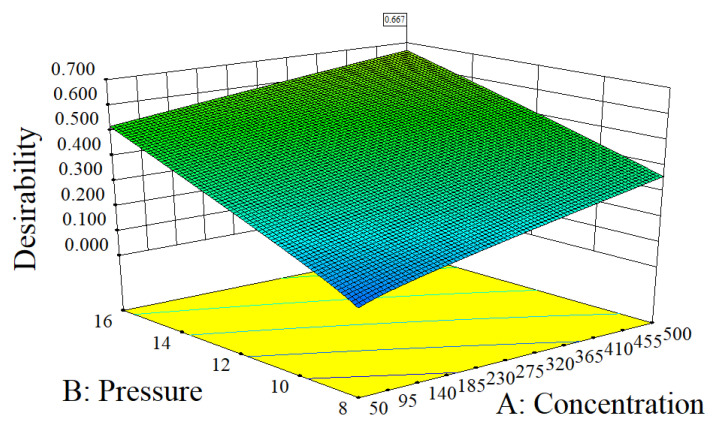
Ideal 3D surface fitting. The concentration range is 50–500 μg/L, the pressure range is 8–16 bar, and the pH range is 4–10.

**Table 1 membranes-13-00355-t001:** Experimental range and level of factors in a factorial experiment.

Factor	Name	Units	Type	Minimum	Maximum	Coded Low	Coded High
A	Concentration	μg/L	Numeric	50	1000	−1 ↔ 50	+1 ↔ 1000
B	Pressure	bar	Numeric	8	20	−1 ↔ 8	+1 ↔ 20
C	pH	/	Numeric	3	11	−1 ↔ 3	+1 ↔ 11

**Table 2 membranes-13-00355-t002:** Factorial design matrix of three variables and the experimental response of PPCP removals.

Run	Factor1 A	Factor2 B	Factor3 C	IBU Removal	CBZ Removal	TCS Removal
1	1000	20	3	99.98	96.51	99.58
2	50	8	11	97.52	95.31	98.10
3	50	8	11	96.89	93.54	98.05
4	1000	20	11	99.76	99.60	99.55
5	1000	8	3	97.90	95.90	99.26
6	50	20	3	99.01	95.81	98.95
7	50	20	3	99.63	97.53	98.99
8	1000	8	11	98.49	98.98	99.23
9	1000	20	3	99.79	98.28	99.63
10	50	20	11	99.60	98.89	98.92
11	50	8	3	97.57	93.90	98.17
12	50	20	11	98.95	97.21	98.87
13	1000	8	11	97.88	97.27	99.19
14	1000	8	3	98.56	97.61	99.30
15	1000	20	11	99.95	97.89	99.51
16	50	8	3	96.91	92.19	98.12

**Table 3 membranes-13-00355-t003:** ANOVA results of PPCPs reverse osmosis removal.

	Source	Sum of Squares	df	Mean Square	*f*-Value	*p*-Value	
IBU removal	Model	16.64	7	2.38	15.04	0.0005	significant
	A-Concentration	2.43	1	2.43	15.40	0.0044	
	B-pressure	14.03	1	14.03	88.79	<0.0001	
	C-pH	0.01	1	0.01	0.04	0.8547	
	AB	0.17	1	0.17	1.07	0.3303	
	AC	0.00	1	0.00	0.00	0.9961	
	BC	0.00	1	0.00	0.00	0.9922	
	ABC	0.00	1	0.00	0.00	0.9786	
	Pure Error	1.26	8	0.16			
	Cor Total	17.91	15				
	Std. dev.	0.40		R^2^		0.9294	
	Mean	98.65		Adjusted R^2^		0.8676	
CBZ removal	Model	54.09	7	7.73	5.20	0.0168	significant
	A-Concentration	19.50	1	19.50	13.14	0.0067	
	B-pressure	18.12	1	18.12	12.20	0.0082	
	C-pH	7.51	1	7.51	5.06	0.0546	
	AB	8.95	1	8.95	6.03	0.0396	
	AC	0.00	1	0.00	0.00	0.9864	
	BC	0.00	1	0.00	0.00	0.9953	
	ABC	0.00	1	0.00	0.00	0.9921	
	Pure Error	11.88	8	1.48			
	Cor Total	65.97	15				
	Std. dev.	1.22		R^2^		0.8199	
	Mean	96.65		Adjusted R^2^		0.6624	
TCS removal	Model	4.73	7	0.68	726.45	<0.0001	significant
	A-Concentration	3.15	1	3.15	3383.12	<0.0001	
	B-pressure	1.31	1	1.31	1409.87	<0.0001	
	C-pH	0.02	1	0.02	22.94	0.0014	
	AB	0.25	1	0.25	269.23	<0.0001	
	AC	0.00	1	0.00	0.00	0.9932	
	BC	0.00	1	0.00	0.00	0.9971	
	ABC	0.00	1	0.00	0.00	0.9937	
	Pure Error	0.01	8	0.00			
	Cor Total	4.74	15				
	Std. dev.	0.03		R^2^		0.9984	
	Mean	98.96		Adjusted R^2^		0.9971	

## Data Availability

The data presented in this study are available on request from the corresponding author.

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
