# Peer review of "Removal of Typical PPCPs by Reverse Osmosis Membranes: Optimization of Treatment Process by Factorial Design"

_membranes, 2023, doi:10.3390/membranes13030355_

Round 1

Reviewer 1 Report

This is a very interesting article. Due to this I agree with the importance of this manuscript.

Regarding the possibilities to improve it, I would like to comment the following if it could help to the authors:

1. To highlight in the abstract the goal of the study, its value and the benefits of the results obtained.

2. To introduce more explications for each graphic presented in the results section.

3. To maintain a similar form of the presentation of the figures shown, when it is possible to look more similar configuration.

4. To show discussions of the results with the references shown.

5. To introduce references of this year 2023. It would be good to increase the number of references with more recent articles.

Thanks so much and good job. I like this very much.

Author Response

(1) We rechecked the manuscript for English language and had it checked by native English-speaking colleague.

(2) We have supplemented the purpose and significance of this study in the abstract.

(3) We explained more about each figure in the results section.

(4) We changed the rendering of the graphics to ensure that the configuration is similar.

(5) We have revised the cited references to ensure that all references are relevant to the manuscript content.

Thank you very much for giving us an opportunity to revise our manuscript.

Reviewer 2 Report

This is a good work with novelty. I recommend minor revision.

1. Figure 5 & 6, please add the units for each parameters.

2.  The significant digit used in this work is inconsistent. 

Author Response

(1) We have added the range and units of each experimental parameter to the figure headings in Figure 5 & 6.

(2) We re-examined the significant numbers in the manuscript and re-calculated the statistical data in the manuscript to ensure that each particular statistical data retained the same number of decimal places to ensure that they were meaningful.

Thank you very much for giving us an opportunity to revise our manuscript.
